# OpenCodeReasoning: Advancing Data Distillation for Competitive Coding

**Wasi Uddin Ahmad,**[*] **Sean Narenthiran,**[*] **Somshubra Majumdar, Aleksander Ficek, Siddhartha Jain, Jocelyn Huang, Vahid Noroozi, Boris Ginsburg**
NVIDIA
Santa Clara, CA 95051, USA
{wasiuddina, snarenthiran, smajumdar, aficek}@nvidia.com
https://huggingface.co/datasets/nvidia/OpenCodeReasoning

## Abstract

Since the advent of reasoning-based large language models, many have found great success from distilling reasoning capabilities into student models. Such techniques have significantly bridged the gap between reasoning and standard LLMs on coding tasks. Despite this, much of the progress on distilling reasoning models remains locked behind proprietary datasets or lacks details on data curation, filtering and subsequent training. To address this, we construct a superior supervised fine-tuning (SFT) dataset that we use to achieve state-of-the-art coding capability results in models of various sizes. Our distilled models use only SFT to achieve 61.8% on Live-CodeBench and 24.6% on CodeContests, surpassing alternatives trained with reinforcement learning. We then perform analysis on the data sources used to construct our dataset, the impact of code execution filtering, and the importance of instruction/solution diversity. We observe that execution filtering negatively affected benchmark accuracy, leading us to prioritize instruction diversity over solution correctness. Finally, we also analyze the token efficiency and reasoning patterns utilized by these models.

## 1   Introduction

Large Language Models (LLMs) have demonstrated a remarkable capacity to excel at a variety of coding capabilities (Hui et al., 2024; Li et al., 2023a; Guo et al., 2024; Roziere et al., 2023; Ahmad et al., 2021). While fine-tuning on code question-solution pairs has led to much of the improved performance, high-quality human-labeled data is limited and expensive to curate. To overcome this bottleneck, many have successfully leveraged LLMs to generate high-quality synthetic code data (Luo et al., 2024; Yu et al., 2024). Notably, works like Wei et al. (2024a) and Huang et al. (2025) have generated diverse instruction-solution pairs, subsequently fine-tuning base models to achieve top results in HumanEval (Chen et al., 2021), MBPP (Austin et al., 2021) and BigCodeBench (Zhuo et al., 2025) benchmarks.

Since the successes of synthetic data generation for code, reasoning-based LLMs have presented the next paradigm in advancing large language model capabilities (Team et al., 2025). Past works such as DeepSeek-AI et al. (2025) have led the way in improving LLM capabilities on reasoning oriented tasks such as math and coding by leveraging large-scale reinforcement learning (Luo et al., 2025) and rule-based reward models. By continuously applying reinforcement learning (RL) with a ground truth verifier on a task such as coding, models learn to apply continuous test-time computation to solve more difficult reasoning-based tasks (Hosseini et al., 2024; Setlur et al., 2025).

Given the improvements from fine-tuning on synthetic data and reasoning capabilities for coding capabilities, many works have found continuous enhancements by combining the two. This has involved distilling the chain-of-thought responses (Wei et al., 2022) from

---

[*] Equal Contribution

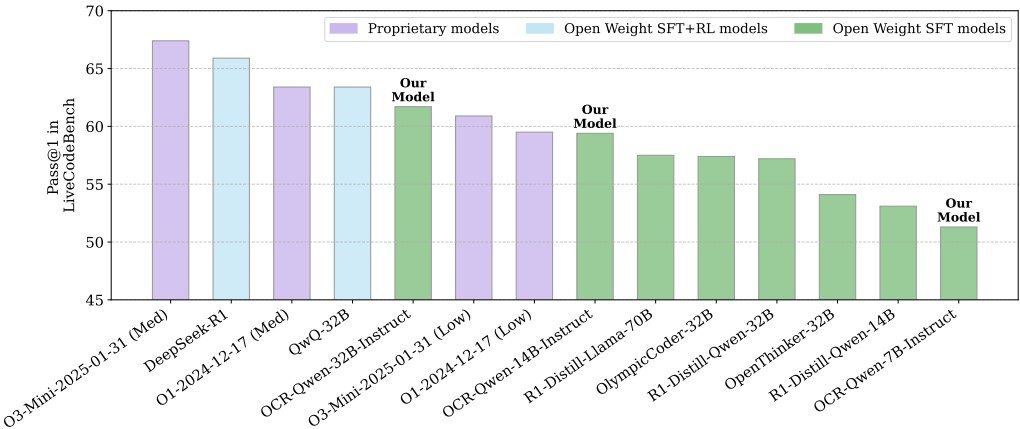

Figure 1: Accuracy (pass@1) comparison between our SFT-only models with proprietary, open-weight SFT-only and SFT+RL models on LiveCodeBench (2408-2502).

better reasoning models to smaller student models by means of supervised fine-tuning (SFT). DeepSeek-R1 successfully distilled R1 reasoning traces, achieving the highest scores on LiveCodeBench (Jain et al., 2025) for medium-sized models: 57.2 (DeepSeek-R1-Distill-Qwen-32B) and 57.5 (DeepSeek-R1-Distill-Llama-70B). Follow up works have found that SFT using only 17k or 114k reasoning samples can be used to improve Qwen2.5-32B-Instruct on math and coding (BespokeLabs, 2025; OpenThoughts, 2025), realizing a 40% gain on AIME 2024 (Patel et al., 2024) and 8.1% gain on LiveCodeBench (Jain et al., 2025).

While distilled models typically rely on SFT with thousands of examples (OpenThoughts, 2025; Team, 2025a; BespokeLabs, 2025), leading open-weight reasoning models achieve superior performance through a training regimen combining SFT and RL (Team, 2025b; Team et al., 2025; DeepSeek-AI et al., 2025), thus maintaining a significant performance advantage over SFT-only models. However, the extent to which SFT can improve reasoning performance is poorly understood, limited by the lack of large-scale reasoning datasets. In an effort to bridge the performance disparity between SFT-only and SFT+RL models, we present OPENCODEREASONING in this work.

OPENCODEREASONING, the largest reasoning-based synthetic dataset to date for coding, comprises 736,712 samples in Python across 28,904 unique competitive programming questions. Fine-tuning Qwen2.5 base and instruct LLMs (7B, 14B, and 32B) with OPEN-CODEREASONING significantly outperformed R1-Distill-Qwen and other open-weight SFT-only models. As illustrated in Figure 1, our SFT-only 7B and 14B models achieved pass@1 rates of 51.3 and 59.4 on LiveCodeBench, respectively, surpassing R1-Distill-Qwen models of the same size by 13.7 and 6.3 absolute points. Furthermore, our 32B model achieved a pass@1 rate of 61.8 (average@64), surpassing two OpenAI models (O1 and O3-Mini) and significantly narrowing the performance gap with DeepSeek-R1, which scored 65.9 pass@1. Beyond the compelling empirical results, we provide a thorough ablation and analysis, providing insights to advance future research.

The contributions of this work can be summarized as follows:

1. We construct and release OPENCODEREASONING, a large-scale dataset of 736,712 DeepSeek-R1 generated code solutions (in Python) with reasoning traces, covering 28,904 unique competitive programming questions, making it the largest dataset of its kind.

2. We validate the efficacy of OPENCODEREASONING by fine-tuning Qwen2.5 models (7B, 14B, and 32B), achieving state-of-the-art performance on LiveCodeBench and CodeContests benchmarks, surpassing similarly sized SFT-only models.

3. We perform an in-depth ablation and analysis to provide insights on execution-based filtering, mixing solutions in multiple languages, reasoning length and patterns of the fine-tuned models. We also provide a replicable recipe for future datasets by detailing our data sourcing and filtering methods.

## 2 OPENCODEREASONING: Dataset Construction and Refinement

This section outlines the construction process of the OPENCODEREASONING dataset, which consists of three steps. First, we collect a diverse set of competitive coding questions from various sources. Second, we utilize a reasoning-enabled large language model (LLM) to generate responses. Third, we post-process these responses, validating reasoning and extracting solution excerpts. To further understand the impact of our construction choices, we conclude with an ablation study, providing valuable insights for future dataset development.

### 2.1 Coding Questions Collection

To construct OPENCODEREASONING, we gathered questions from TACO (Li et al., 2023b), APPS (Hendrycks et al., 2021), CodeContests (Li et al., 2022), and CodeForces from the OpenR1 project (Penedo et al., 2025a). Given the limited availability of unique competitive coding problems and the overlapping nature of public datasets, we performed exact-match deduplication, resulting in 28,904 distinct questions across a range of difficulties. Table 1 displays the individual breakdown of the source of the questions in that dataset.

| Source | # Question | # Sample |
|---|---|---|
| AIZU | 2151 | 62,476 |
| AtCoder | 2080 | 47,222 |
| CodeChef | 3869 | 72,925 |
| CodeForces | 10403 | 388,405 |
| Codewars | 2515 | 34,326 |
| GeeksForGeeks | 2674 | 37,602 |
| HackerEarth | 2285 | 59,181 |
| HackerRank | 914 | 10,955 |
| Kattis | 1236 | 13,095 |
| LeetCode | 777 | 10,525 |
| Total | 28,904 | 736,712 |

Table 1: OpenCodeReasoning statistics.

**Verification for Benchmark Contamination**  We meticulously check for any potential data leakage or overlap between the collected coding questions and evaluation benchmarks (Jain et al., 2025; Li et al., 2022; Chen et al., 2021; Austin et al., 2021). We follow the procedure described by Yang et al. (2023), calculating cosine similarity (threshold 0.7) to find the nearest neighbor in the benchmarks for each unique question in our dataset. We then employed Llama-3.3-70B-Instruct (Grattafiori et al., 2024) and Qwen2.5-32B-Instruct (Qwen, 2024) as judges to assess the semantic similarity of these pairs. Manual inspection of the 90 potentially problematic samples (representing $\leq 0.3\%$ of our dataset) confirmed that they were not paraphrases or semantically similar. Consequently, we proceeded to the solution generation step for the entire set of 28,904 questions.

### 2.2 Solution Code Generation

In this step, we generate multiple solutions per question using the DeepSeek-R1 (DeepSeek-AI et al., 2025). We primarily generate solutions in Python programming language. Additionally, we generate solutions in C++ language to perform preliminary experiments on the harder IOI benchmark (Penedo et al., 2025b). All solutions are sampled via Nucleus Sampling (Holtzman et al., 2020), using temperature 0.6, top-p 0.95, and explicitly injecting *<think>* tag to force the model to generate reasoning traces. We use SGLang (Zheng et al., 2024) for R1 generations with a maximum output sequence length of 16k tokens.

### 2.3 Post-Processing for Refinement

We refine the synthesized solutions to improve the quality of OPENCODEREASONING while balancing the diversity of instructions. First, we verify whether the solution includes reasoning traces enclosed within *<think>* and *</think>* tags. Subsequently, we extract the solution segments, isolating the reasoning traces from the responses. We then verify the presence of a code block within the solution segments, specifically delimited by either ```python ... ``` or ```cpp ... ```. We filter out responses where the generated reasoning traces contain code blocks, simplifying the evaluation of benchmarks. This filtering process resulted in the removal of a remarkably small number of responses. Furthermore, we verify the syntactic correctness of the solution code blocks by parsing them using Tree Sitter (TreeSitter, 2013). The refinement step yielded a total of 736,712 Python samples (detailed breakdown in Table 1) and 355,792 C++ samples. A detailed breakdown of token count for the Python samples is shown in Figure 2.

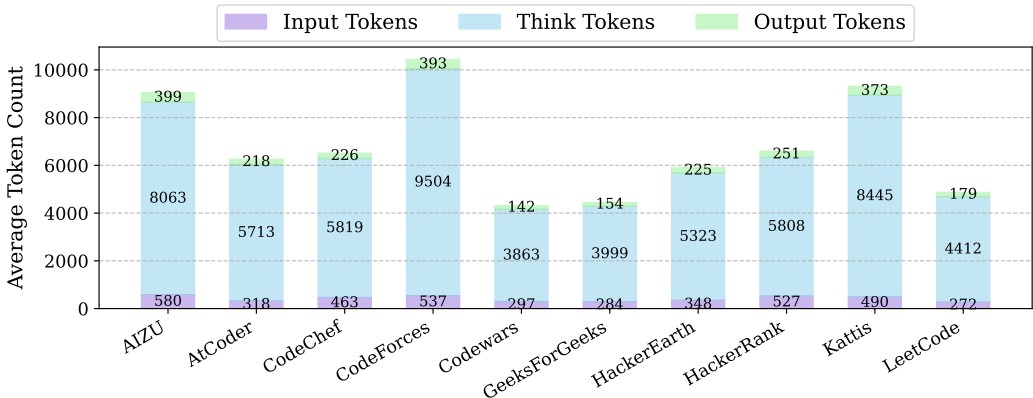

Figure 2: An illustration showing the average number of tokens (tokenizing using Qwen2.5 tokenizer (Yang et al., 2024)) per Python sample across dataset sources.

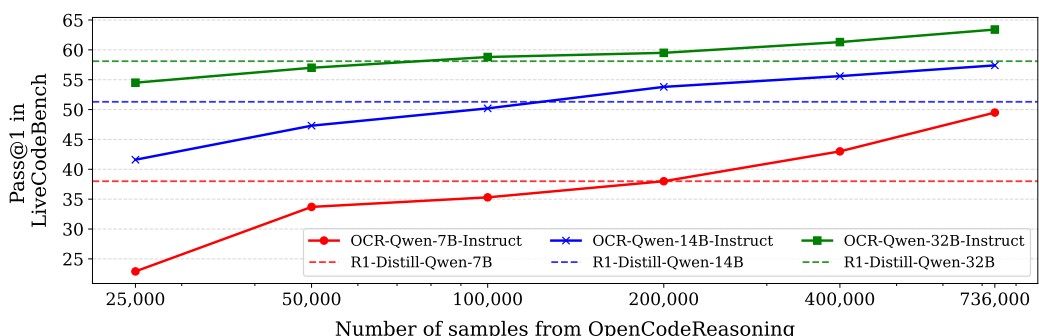

Figure 3: Impact of scaling up data from 25k to 736k samples in OPENCODEREASONING.

## 2.4 Scaling Up Data in Stages

Several works in recent literature have shown that a minuscule amount of SFT data can induce reasoning capabilities in LLMs (HuggingFace, 2025; Muennighoff et al., 2025; BespokeLabs, 2025; OpenThoughts, 2025). However, they primarily focus on the domain of advanced mathematics, and evaluation on code benchmarks is seldom done. Contrary to their findings, we find that while reasoning ability itself may be induced with a minute amount of data, to achieve state-of-the-art results on coding benchmarks, large datasets are necessary. Therefore, we investigated the impact of data scaling on model performance, demonstrating a positive correlation between increased training data and improved results.

We incrementally expand our dataset from 25k to 736k samples in stages, which can be seen in Figure 3. Initially, we utilized only the 13k questions from CodeContests and generated a number of solutions for each question using DeepSeek-R1. Notably, the initial scaling from 25k to 100k samples yielded substantial gains on the LiveCodeBench benchmark. We then performed an exploratory analysis of the solutions generated by the model and found that there is a significant imbalance in the number of successful samples generated for easy and medium difficulty problems versus hard competition problems. For example, R1 cannot successfully generate solutions that pass more than 40% of the hard tasks in LiveCodeBench. As such, we select just the hard subset of questions from the CodeContests train set, which resulted in roughly 4.5k instructions, and generate multiple solutions for them. Finally, we integrate additional instructions to compose the final 28k unique question set, and generate solutions using DeepSeek-R1 for the new questions, resulting in a final collection of 736,712 samples. This final expansion produced the most significant improvements, underscoring the benefits of large-scale training data across diverse problem types.

It is to be noted that the scaling curve in Figure 3 does not plateau despite the use of 736k samples, and it remains to be seen when the model's improvement will saturate and yield diminishing returns. The most significant gains in benchmark scores were observed by simply increasing the number of unique, varied, and difficult questions. This suggests the research community may need to find methods to construct or mine problems of sufficient difficulty at a larger scale to bolster results by a significant margin.

# 3 Main Evaluation

**Training and Inference Hyper-Parameters**  We assessed the effectiveness of SFT using OPENCODEREASONING by fine-tuning the Qwen2.5 base and instruct models at 7B, 14B, and 32B parameters. The models were trained for 3 epochs on NVIDIA H100-80GB GPUs, using the AdamW optimizer (Kingma & Ba, 2015) with a batch size of 256 and a maximum sequence length of 32,768. An initial learning rate grid search, covering $[1e-5, 3e-5, 5e-5, 8e-5, 1e-4]$, identified $5e-5$ as optimal across model sizes. We employed a CosineAnnealing scheduler with a warmup ratio of 0.1, and used the final checkpoint for evaluation. Training acceleration was achieved through sequence packing (Shen et al., 2024), tensor and context parallelism, and BF16 precision. For inference, we employed temperature-based nucleus sampling (Holtzman et al., 2020). An initial temperature sweep over the values $[0.0, 0.2, 0.6, 0.7, 1.0]$ revealed that temperatures of 0.6 and 0.7 yielded the best performance. Consequently, we selected a temperature of 0.6 for all subsequent experiments. Inference was performed using vLLM (Kwon et al., 2023) with a maximum generation length of 30,720 tokens.

**Evaluation Benchmarks and Baselines**  We evaluate our fine-tuned models and the baseline models on LiveCodeBench (Jain et al., 2025) and CodeContests (Li et al., 2022). For this study, we utilized LiveCodeBench problems within the date range of 2408 to 2502, which encompassed a total of 279 coding problems. To mitigate performance variance inherent in single-run evaluations, we report the average pass@1 metric, calculated by averaging 64 inference runs for LiveCodeBench and 16 runs for CodeContests. The following open-weight models were utilized as baselines in our evaluation: DeepSeek-R1 and R1-Distill-Qwen models (DeepSeek-AI et al., 2025), QwQ-32B (Team, 2025b), OlympicCoder (Penedo et al., 2025a), Bespoke-Stratos (BespokeLabs, 2025), and OpenThinker (OpenThoughts, 2025).

## 3.1 Main Results

Table 2 summarizes the performance of our distilled OCR-Qwen models compared to a variety of competing baselines. We consistently observed the following three trends.

**Competitive Scores at Small Scales**  Within the 7B parameter model category, both OCR-Qwen-7B and OCR-Qwen-7B-Instruct demonstrated superior performance, with the instruct model significantly surpassing the best-performing baseline, OlympicCoder-7B, across both benchmarks (**10.4**% and **7.5**% absolute improvements on LiveCodeBench and CodeContests, respectively). The results underscore the effectiveness of our distillation approach in transferring reasoning capabilities, even at lower parameter counts.

**Scaling Yields Rapid Gains**  Moving from 7B to 14B and 32B reveals pronounced improvements across all models, especially for the OCR-Qwen family. In the 14B range, OCR-Qwen-14B-Instruct attains a **59.4** average pass@1 on LiveCodeBench and a **23.6** pass@1 on CodeContests, edging out other 14B distilled models. These results indicate that the OPENCODEREASONING is highly effective in training models across a range of model scales.

**Narrowing the Gap to Top-Tier Models**  At the 32B scale, OCR-Qwen-32B and OCR-Qwen-32B-Instruct outperformed strong competitors like R1-Distill-Qwen-32B and OlympicCoder-32B. OCR-Qwen-32B achieved the highest pass@1 scores among our fine-tuned models, surpassing QwQ-32B with **61.8** on LiveCodeBench and **24.6** on CodeContests. Notably, these scores are only slightly lower than DeepSeek-R1's 65.6 and 26.2, respectively, showcasing the strong competitive performance of OCR-Qwen models at the 32B scale.

| Model | LiveCodeBench | | | | CodeContest | | | |
|---|---|---|---|---|---|---|---|---|
| | Easy | Medium | Hard | Avg. | Public | Private | Generated | All |
| DeepSeek-R1 | 98.5 | 79.8 | 37.4 | 65.6 | 61.9 | 38.4 | 44.3 | 26.2 |
| QwQ-32B | 97.0 | 79.8 | 28.5 | 61.3 | 52.9 | 32.2 | 34.2 | 20.2 |
| **Distilled 7B+ Models** | | | | | | | | |
| Bespoke-Stratos-7B | 49.3 | 9.0 | 0 | 14.7 | 4.2 | 4.6 | 3.0 | 2.0 |
| OpenThinker-7B | 80.6 | 16.9 | 1.6 | 25.5 | 11.7 | 9.0 | 7.8 | 5.0 |
| R1-Distill-Qwen-7B | 86.6 | 43.8 | 7.0 | 38.0 | 26.7 | 17.6 | 19.5 | 11.1 |
| OlympicCoder-7B | 82.1 | 49.4 | 12.2 | 40.9 | 30.3 | 19.9 | 19.0 | 10.6 |
| OCR-Qwen-7B | 92.5 | 61.4 | 15.2 | 48.5 | 42.5 | 27.1 | 29.5 | 16.3 |
| OCR-Qwen-7B-Instruct | 95.4 | 64.0 | 18.0 | **51.3** | **46.7** | **29.6** | **32.3** | **18.1** |
| **Distilled 14B+ Models** | | | | | | | | |
| R1-Distill-Qwen-14B | 98.5 | 62.9 | 17.1 | 51.3 | 44.0 | 29.1 | 31.7 | 17.6 |
| OCR-Qwen-14B | 97.0 | 71.4 | 26.3 | 57.7 | **57.4** | 34.3 | 39.6 | 22.6 |
| OCR-Qwen-14B-Instruct | 97.6 | 74.4 | 27.6 | **59.4** | 57.1 | **34.5** | **40.2** | **23.6** |
| **Distilled 32B+ Models** | | | | | | | | |
| Bespoke-Stratos-32B | 80.6 | 30.3 | 2.4 | 30.1 | 17.3 | 14.8 | 11.1 | 6.3 |
| OpenThinker-32B | 97.0 | 65.2 | 22.8 | 54.1 | 41.1 | 25.7 | 28.7 | 16.4 |
| R1-Distill-Qwen-32B | 98.5 | 68.5 | 28.5 | 58.1 | 45.9 | 30.6 | 32.5 | 18.3 |
| OlympicCoder-32B | 98.5 | 71.9 | 24.4 | 57.4 | 45.5 | 29.3 | 30.5 | 18.0 |
| OCR-Qwen-32B | 98.2 | 76.2 | 31.5 | **61.8** | 59.8 | **36.8** | 42.1 | **24.6** |
| OCR-Qwen-32B-Instruct | 98.4 | 77.2 | 30.4 | 61.7 | **60.3** | 36.6 | **42.7** | 24.4 |

Table 2: Performance comparison of *open-weight* reasoning models on LiveCodeBench and CodeContest. Highlighted rows show our finetuned models' performances, averaged over 64 inference runs. Baselines were run once. Bold indicates the highest performance.

## 4 Ablation and Analyses

### 4.1 Ablation: Filtering by Code Execution

Despite DeepSeek-R1's strong code generation capabilities, its outputs may contain errors, failing to pass unit tests. To assess the impact of these erroneous solutions on fine-tuned model performance, we conducted an ablation study using a subset of OPENCODEREASONING (includes 445k samples) derived from the CodeContest subset, which provides unit tests to assess solution correctness.[1]

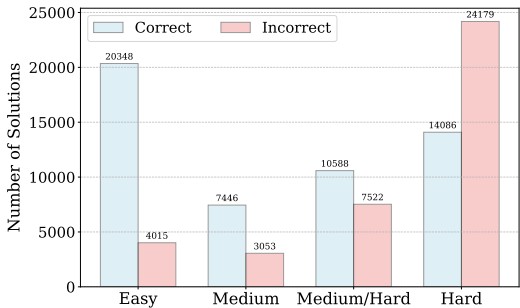

Figure 4: Distribution of correct vs. incorrect samples in the CodeContest subset of OPEN-CODEREASONING across four difficulty levels, as determined by unit tests.

We fine-tuned Qwen-2.5-14B-Instruct using three subsamples - (1) the full subsample of 445k instances; (2) selecting all instances that pass unit-tests (151k instances); and (3) selecting an equal number of samples as in (2) that fail all tests. Surprisingly, we observed that fine-tuning on incorrect solutions results in higher accuracy than on correct solutions. The results are shown in Table 3. Investigating further, we identified that incorrect solutions span questions that are more challenging than the ones associated with the correct solutions. To visualize it, we provide the correct and incorrect sample distribution across

---

[1]The test suites are categorized into "Public", "Private" and "Generated" sets. The "Generated" tests are derived from ground-truth solutions to enhance coverage, which "Public" and "Private" tests may not be provided. Given an average of 192.7 generated tests per problem and a 445k sample dataset, we limited tests to 50 to reduce computational cost.

| Filtering | Dataset Size | LiveCodeBench | CodeContests |
|---|---|---|---|
| No Filtering | 445,618 | 54.1 | 16.59 |
| Correct Solutions | 151,251 | 47.0 | 15.34 |
| Incorrect Solutions | 151,251 | 52.3 | 15.53 |

Table 3: Evaluation results of finetuning Qwen2.5-14B-Instruct on execution filtered subsample of OPENCODEREASONING, pertaining to CodeContest training dataset.

difficulty levels in Figure 4. This suggests that, despite the generation of incorrect solutions by large teacher models for challenging problems, positive transfer can still occur through distillation. We consider this a promising avenue for future research.

## 4.2 Ablation: Inclusion of C++ Solutions

| Model | Dataset Size | | LiveCodeBench (pass@1) | CodeContests (pass@1) | IOI (Total Score) |
|---|---|---|---|---|---|
| | Python | C++ | | | |
| OlympicCoder-7B | 0 | 100K | 40.9 | 10.6 | 127 |
| OlympicCoder-32B | 0 | 100K | 57.4 | 18.0 | 153.5 |
| QWQ-32B | - | - | 61.3 | 20.2 | 175.5 |
| OCR-Qwen-14B-Instruct | 736K | 0 | 59.4 | 23.6 | 145.5 |
| | 0 | 356K | 54.2 | 18.0 | 153.5 |
| | 736K | 356K | 59.7 | 23.4 | 153.5 |
| OCR-Qwen-32B-Instruct | 736K | 0 | 61.7 | 24.4 | 145.5 |
| | 0 | 356K | 60.6 | 21.0 | 168.5 |
| | 736K | 356K | 61.5 | 25.5 | 175.5 |

Table 4: Evaluation results with inclusion of R1 generated samples in C++ languages with OPENCODEREASONING. For IOI, we select the most common score out of 8 runs.

While OPENCODEREASONING primarily consists of Python samples, we sought to investigate whether incorporating solutions in other languages could enhance distillation. Consequently, we explored the inclusion of 356k C++ samples, generated by R1, in conjunction with OPENCODEREASONING. In addition to LiveCodeBench and CodeContest, we evaluated on the IOI benchmark (Penedo et al., 2025b), which requires C++ solutions. The results are presented in Table 4. It is evident that the inclusion of C++ solutions has no positive impact on Python benchmark performance, but does significantly improve the accuracy on the C++ benchmark. Finding an optimal strategy to effectively leverage multilingual data warrants further research.

## 4.3 Analysis: How long does an LLM think before generating code solution?

To investigate the reasoning process of LLMs before code generation, we analyzed the average token count produced by DeepSeek-R1, QwQ-32B, and our fine-tuned OCR-32B models during their thinking phase. We posit that more difficult problems would elicit longer reasoning traces. To test this hypothesis, we evaluated the models on LiveCodeBench, which categorizes problems by difficulty (easy, medium, hard). We conducted inference with a maximum generation length of 32k tokens. The results, presented in Figure 5, reveal several notable trends that we discuss below.

We observe a stark difference in the number of tokens generated for easy vs. medium vs. hard problems, a trait shared by all models. While DeepSeek-AI et al. (2025) corroborates this observation, we find that OCR-32B models closely follow the token budget of the significantly larger DeepSeek-R1 model. Notably, QwQ-32B requires significantly more tokens than other models to achieve comparable scores, particularly evident when evaluated on *hard* problems. In comparison, OCR-32B models can obtain similar results using 20-30% fewer tokens across all difficulty levels.

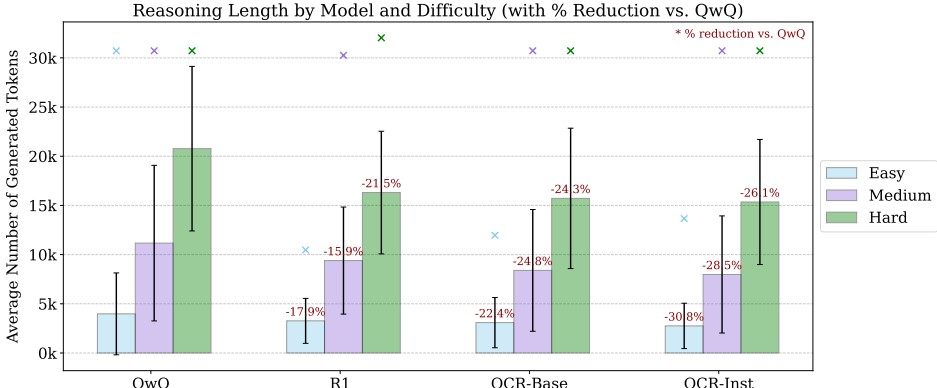

Figure 5: Comparison of reasoning (thinking) lengths, partitioned by difficulty of the task.

During evaluation, we observed that a significantly larger token budget is required for *hard* problems, as models often did not complete their reasoning phase within 16k tokens. However, extending this budget to 32k tokens did not translate to improved accuracy. In fact, no substantial performance gains were observed on the *hard* problems. Our findings corroborate Wang et al. (2025) that suggests LLMs tend to be incorrect with more tokens, possibly due to increased backtracking in incorrect solutions (more analysis in the next section). Finally, as a consequence of long reasoning traces, models sometimes enter an unrecoverable *reasoning loop*, consuming all available tokens while repeating reasoning steps instead of producing a solution. This behavior was observed to a limited extent across all models despite a large token budget of 32k.

## 4.4 Analysis: Is there any pattern in LLMs' reasoning traces?

Following Gandhi et al. (2025), to gain insights into the problem-solving strategies of LLMs with reasoning capabilities, we analyzed the prevalence of various patterns in the generated reasoning traces across problems with different difficulty levels (easy, medium, hard) and their correlation with solution correctness. Detailed description of how we collected the reasoning patterns is provided in the Appendix.

| Reasoning pattern | Correct solutions | | Incorrect solutions | |
|---|---|---|---|---|
| | $E \rightarrow M$ | $M \rightarrow H$ | $E \rightarrow M$ | $M \rightarrow H$ |
| **Backtracking** | 0.05 → 0.06 | 0.06 → 0.08 | 0.08 → 0.07 | 0.07 → 0.08 |
| **New Idea Generation** | 0.16 → 0.20 | 0.20 → 0.23 | 0.16 → 0.22 | 0.22 → 0.25 |
| **Problem Rephrasing** | 0.21 → 0.20 | 0.20 → 0.20 | 0.21 → 0.22 | 0.22 → 0.23 |
| **Self-Evaluation** | 0.39 → 0.37 | 0.37 → 0.34 | 0.36 → 0.34 | 0.34 → 0.31 |
| **Solving a Simpler Problem First** | 0.06 → 0.05 | 0.05 → 0.05 | 0.06 → 0.05 | 0.05 → 0.03 |
| **Subgoal Generation** | 0.13 → 0.12 | 0.12 → 0.10 | 0.13 → 0.10 | 0.10 → 0.10 |

Table 5: Demonstrating how the frequency of a particular reasoning pattern changes as problem difficulty increases. Changes highlighted in red or green are statistically significant (p-value threshold of 0.05) with red indicating a decline in frequency and green an increase. E/M/H correspond to the Easy/Medium/Hard ratings for problems.

We began by analyzing the change in reasoning pattern proportions across problem difficulty levels (easy, medium, hard) for both correct and incorrect solutions, as detailed in Table 5. Our findings reveal a dynamic adjustment of reasoning strategies in response to problem complexity. Both correct and incorrect solutions showed an increase in *exploration-related* patterns. Correct solutions, however, also demonstrated an increase in *backtracking*. Incorrect solutions maintained elevated *backtracking* levels across all difficulties, indicating that LLMs recognize errors but struggle to correct them. This indicates a heightened demand for exploratory reasoning as problem difficulty escalates or when models pursue incorrect

initial reasoning paths. Furthermore, we observed a decline in *self-evaluation* proportions, likely due to the increased presence of exploratory patterns.

To understand the reasoning patterns that contribute to solution correctness, we conducted a paired comparison for each problem, examining the proportion of specific patterns between correct and incorrect generations. As shown in Table 6, while most reasoning patterns did not exhibit statistically significant differences, *self-evaluation* and *subgoal* generation emerged as notable exceptions. Both were significantly more prevalent in correct solutions, indicating that self-reflection and problem decomposition are crucial for accuracy. Furthermore, to quantify the diversity of reasoning patterns, we computed entropy for both correct

| Reasoning pattern | Fraction |
|---|---|
| **Backtracking** | 0.49 |
| **New Idea Generation** | 0.48 |
| **Problem Rephrasing** | 0.44 |
| **Self-Evaluation** | 0.58 |
| **Solving a Simpler Problem First** | 0.55 |
| **Subgoal Generation** | 0.58 |

Table 6: Results showing the fraction of problems for which a particular reasoning pattern is more frequent in the correct solutions vs the incorrect solutions. Numbers highlighted in green are statistically significant.

(1.26) and incorrect generations (1.19). The results suggest that employing diverse reasoning strategies improves solution correctness.

## 5 Related Works

This research builds upon a body of work exploring both reasoning capabilities in large language models (LLMs) and the generation of synthetic code datasets, with a particular focus on their intersection. Early investigations into eliciting reasoning, such as Wang et al. (2023a)'s Chain-of-Thought prompting, laid the groundwork for subsequent fine-tuning approaches using generated reasoning traces (Magister et al., 2023). More recently, reinforcement learning has been shown to significantly enhance LLM reasoning performance (DeepSeek-AI et al., 2025; Xiang et al., 2025). We leverage these advancements to create a synthetic dataset capable of distilling reasoning capabilities into smaller models.

Concurrent to these developments, researchers have explored methods for generating synthetic datasets for code. Initial approaches used LLMs to create solutions and diversify problems (Wang et al., 2023b; Liu et al., 2024). This was followed by techniques that augmented prompts to refine LLM-generated solutions for improved code generation (Xu et al., 2024; Majumdar et al., 2024; Wei et al., 2024a). Additionally, some studies have focused on generating question-solution pairs by using scraped code blocks to produce corresponding questions (Wei et al., 2024b; Wu et al., 2024).

Recently, datasets containing 17k to 114k reasoning-based question-solution pairs have been released, demonstrating improved performance of fine-tuned models on coding benchmarks, often presented via blog posts (BespokeLabs, 2025; Penedo et al., 2025c; OpenThoughts, 2025). Li et al. (2025) examined the effects of distilling and fine-tuning with 17k long CoT reasoning solutions. Xu et al. (2025) generated a dataset comprising 447K samples with DeepSeek-R1 reasoning traces; however, their fine-tuning focused only on smaller subsets validated through LLM-based tests. Unlike these approaches, our work investigates the impact of scaling synthetic data to 736,712 samples, demonstrating that larger datasets result in state-of-the-art performance for supervised fine-tuned models across various model sizes.

## 6 Conclusion

We present OPENCODEREASONING, the largest instruction tuning dataset to date for code generation with reasoning. Fine-tuning Qwen2.5 base and instruct models across various sizes (7B, 14B, and 32B) using this dataset significantly outperforms DeepSeek-R1-Distill-Qwen models on LiveCodeBench and Code-Contests benchmarks. We also provide research insights by performing ablation and in-depth analysis, demonstrating the impact of different considerations for fine-tuning. The OPENCODEREASONING dataset will be fully open-sourced to advance LLM-for-code-reasoning research.

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

# Supplementary Material: Appendices

## A    Reasoning pattern extraction

---

**Initial chain of thought segmentation prompt**

Below is a chain of thought for solving a question. Figure out what are the different reasoning patterns that are used like problem rephrasing, new idea generation, self-evaluation, verification, backtracking, subgoal generation, solving a simpler problem first, and more. Then your task is to segment the entire chain of thought into different reasoning patterns. Rewrite the chain of thought in the following format:

```
<pattern> pattern name </pattern>
<content> the entire text that corresponds to the pattern </content>
```

Chain of thought: {thoughts}

---

Figure 6: Prompt template for segmenting chain of thought into reasoning patterns.

---

**Final chain of thought segmentation prompt**

Below is a chain of thought for solving a question. For the segment between the <unannotated> and < /unannotated> tags, figure out what is reasoning pattern used in that segment like problem rephrasing, new idea generation, self-evaluation, verification, backtracking, subgoal generation, solving a simpler problem first, or something else. Then your task is to identify the reasoning pattern used in the unannotated segment. Rewrite the unannotated segment in the following format:

```
<content> The text within the unannotated segment that corresponds to
the pattern. </content>
<reasoning> Reasoning for what the pattern should be for the content </reasoning>
<pattern> *single* pattern name </pattern>
```

Chain of thought: {thoughts}

---

Figure 7: Prompt template for further segmenting chain of thought into reasoning patterns.

We use Qwen-32B-Instruct model to do the initial segmentation of the chain of thought using the prompt showed in Figure 6. Then for each unannotated segment, we use the prompt in Figure 7 to do one more round of segmentation. The reasoning patterns are then extracted from within the <pattern> tags. We merge the verification and reasoning pattern into self-evaluation itself. Anytime the model labels a segment with multiple patterns, we exclude that segment as we consider the model unsure of what pattern it corresponds to. For each generation we compute the fraction of times a particular pattern appears in the generation out of all the patterns in the generation. Thus for each generation we obtain a vector with element corresponding to how frequently that pattern appeared in the generation. For the analysis of patterns with respect to hardness, we computed the mean of this distribution across all correct and incorrect generations separately, using a t-test to compute significance. To analyze per-problem pattern prevalence, we generated a binary matrix where each row represented a problem and each element indicated whether a specific pattern was more frequent in correct solutions (1) or incorrect solutions (0). Statistical significance was then determined using the binomial test.

