# OpenReview forum: "OpenCodeReasoning: Advancing Data Distillation for Competitive Coding"
_colmweb.org/COLM/2025/Conference — COLM 2025_

### Official Review · Reviewer_2zW4 · 2025-05-08

**Rating:** 5
**Confidence:** 5
**Ethics Flag:** 1

**Summary:**

This paper presents (1) a dataset of solutions to programming problems, including reasoning traces, produced by DeepSeek R; (2) fine-tuned QwenCoder on this dataset to achieve SOTA on LiveCodeBench and CodeContests for an open-weight model; and (3) conducts some ablations.

The most surprising finding is that fine-tuning on *incorrect* solutions improves performance slightly on LiveCodeBench.

**Reasons To Accept:**

1. This paper is a potentially valuable community resource. As far as I know, it is the largest collection of reasoning traces and solutions to programming problems to date. I can easily imagine that it may be used to train many other models in the future.

2. The paper presents a SOTA open-weight model on two benchmarks, which is an achievement, but unlikely a good contribution. We can expect open models to cease being SOTA in weeks. Moreover, they are fine-tuned exclusively on programming contest solutions. There is no evidence that the model will be useful on programming problems that our outside the competitive programming domain. E.g., no evaluation on BigCodeBench, DS-1000, or other non-competition benchmarks.

**Reasons To Reject:**

The primary contribution of this paper is running a large-scale inference job that produced 700K reasoning traces from DeepSeek R1. As mentioned above, this is a valuable community resource, and likely to be a useful contribution that people should know about. But, this work feels like a workshop paper and not a COLM paper to me for the following reasons.

1. The training and evaluation only cover programming contest problems, and not real-world problems.
2. The dataset of prompts are from existing, well-structured data sources. The paper does not perform any significant new data collection effort, other than saving the output of DeepSeek R1.
3. There is limited insight in the ablations. The one surprise, which is a brief note, is that fine-tuning on incorrect solutions seems to give a slightly higher LiveCodeBench score.

---

> ### Author Response · Authors · 2025-05-30
> **[Author response] addressing questions and concerns**
>
> We thank the reviewer for providing feedback on our work. We discuss the concerns below.
>
> **Focus on Competitive Coding**
>
> As our paper title indicates, this paper focuses on competitive coding. There are a significant amount of works in the literature that focus solely on competitive coding capabilities focusing on benchmarks such as HumanEval and MBPP. Therefore, if a particular focus on competitive coding is considered a limitation of our work, then we happily accept it.
>
> **Dataset Contribution**
>
> While it's true that we gathered programming questions from publicly available sources, to the best of our knowledge, this dataset stands out as the largest in terms of unique coding questions, totaling 28,900. Given that DeepSeek-R1 released distilled models without their finetuning corpora, we wanted to contribute a dataset to the community that could be leveraged for Supervised Fine-Tuning (SFT) and enable models to surpass their competitive coding scores.
>
> **Ablation and Analysis**
>
> Through our ablation studies and analysis, we aimed to highlight the impact of data scaling, execution-based filtering, joint training on Python and C++ solutions, reasoning length and patterns of OCR models, among other factors. If the reviewer has any other specific ablation or analysis in mind, please let us know. We'd be happy to include them to improve our work.
>
> ---
>
> Please let us know if you have any other questions or concerns that might change your view of our work.

---

> > ### Comment · Reviewer_2zW4 · 2025-06-10
> >
> > My apologies for the delay. I have read the response. I don't have a specific suggestion on what kind of analysis or ablation to run -- that depends on insights into the dataset that only the authors can have.
> >
> > I will be keeping my score the same.

---

> ### Author Response · Authors · 2025-06-06
> **Seeking further feedback**
>
> Dear Reviewer,
>
> We addressed the concerns you raised during the rebuttal phase. If anything remains unclear, please let us know—we would be glad to provide additional information or run quick experiments before the discussion period ends. If our clarifications resolve your concerns, we would greatly appreciate your reconsideration of the current score.
>
> Thank you again for your thoughtful feedback.

---

> ### Author Response · Authors · 2025-06-10
>
> Dear Reviewer 2zW408,
>
> The discussion period is ending today, and we have not yet received your thoughts on our responses to your comments. If our clarifications have resolved your concerns, would you be willing to reconsider your score in favor of acceptance? Should any questions remain, please let us know—we would be happy to provide additional details right away.
>
> Thank you again for your time and constructive feedback.

---

### Official Review · Reviewer_xjD7 · 2025-05-11

**Rating:** 7
**Confidence:** 5
**Ethics Flag:** 1

**Summary:**

The authors introduce OpenCodeReasoning, a new large-scale and synthetic SFT dataset for code synthesis on competitive programming problems. This work updates, scales-up, and builds upon previous works on synthetic SFT data generation for code like Magicoder and WizardCoder by generating demonstration data from a reasoning LLM (DeepSeek R1) rather than ChatGPT/GPT-4. Fine-tuning LMs on OpenCodeReasoning improves model performance on benchmarks compared to existing R1 distillation datasets, with larger margins of improvement smaller LMs. This is a contribution to the open-source community.

**Questions To Authors:**

- Figure 2 was initially very confusing to me. Do "output tokens" here represent all thinking and answer tokens? Nearly ~10K tokens per final answer on these benchmarks seems like an analysis error -- to my knowledge, answer/final output lengths should still be on the order of a few hundred tokens on average for these data sources.
- Will the full data preparation pipeline also be open-sourced?

**Reasons To Accept:**

- **Well-written paper with thorough experiments, evaluations, and ablations**.
- Open-source research on code synthesis is data-bottlenecked. A new and updated dataset of this scale with careful filtering, de-duplication, and contamination checks is **valuable for the community**.
- Some **interesting and surprising insights on synthetic data generation for competitive coding**. For example, the authors show that "fine-tuning on all incorrect solutions results in higher accuracy than on correct solutions" during distillation, demonstrating that covering difficult examples during SFT may be more important than correctness.

**Reasons To Reject:**

- **Over-claiming and absence of error bars**. For example, in the conclusion, the authors write: "using this dataset significantly outperforms DeepSeek-R1-Distill-32 Qwen models on LiveCodeBench and Code-Contests benchmarks." For 32B models, average improvements over R1-Distill-Qwen-32B are ~2 points on LiveCodeBench, judging by Table 2 and Fig 1. Error bars ala [1] should be added to Table 2, Fig 1, and Fig 3 to support this claim and clarify strength of results.
- **Slight inconsistency in reported scores between figures/tables** Forgive me if I missed something, but the values in Figure 3 do not seem to align exactly with Table 2 and Figure 1? Why is this? E.g. the performance of OCR-Qwen2.5-32B-Instruct on the full OCR dataset seems higher than reported in Fig 1 and Table 2 by a few points (~64% pass@1 vs ~61.8) .


[1] Miller, E. (2024). Adding error bars to evals: A statistical approach to language model evaluations. arXiv preprint arXiv:2411.00640.

---

> ### Author Response · Authors · 2025-05-30
> **[Author response] addressing questions and concerns**
>
> We appreciate the reviewer's feedback on our work and are happy to address the questions and concerns raised.
>
> **Clarifying Performance Reporting and Error Bars**
>
> We appreciate the suggestion to add error bars and will include them in the camera-ready version. We want to assure the reviewer that we have not overclaimed our performances. We reported average scores over 64 runs, consistent with suggestions from publicly available models. For OCR-Qwen-32B, we reported an average pass@1 score of 61.8 across 64 runs. The minimum and maximum pass@1 scores within these 64 runs were 58.8 and 65.2, respectively. Both Figure 1 and Table 2 reflect this 61.8 pass@1 (average@64).
>
> However, in Figure 3, we reported scores from a single run where we achieved 63.5 pass@1. This was done for consistency, as we used a single-run evaluation for checkpoints trained with 25k, 50k, 100k, 200k, and 400k samples.
>
> **Baseline Model Reporting**
>
> For baseline models like R1-Distill-Qwen-32B, we used their publicly available scores in both Figure 1 and Table 2. We also ran these baseline models 16 times ourselves and found their average scores to be very close to the publicly reported numbers, which is why we opted to use the reported accuracies.
>
> This wasn't the case for QwQ-32B. The average scores we obtained for this model in Table 2 (61.3 pass@1) were lower than the 63.4 pass@1 reported in Figure 1. We will clarify this discrepancy further in the camera-ready version of the paper.
>
> **"Output" Tokens and Data Pipeline**
>
> You're absolutely right about "output" tokens encompassing both thinking and answer tokens. We realize now that this wasn't clearly defined in the paper and will modify the details in the camera-ready version for clarity.
>
> Finally, our data preparation pipeline components are already publicly available. We'll provide comprehensive documentation to enable public reproduction of our data generation steps.
>
> ---
>
> We're happy to answer any further questions or concerns that could help you see our work in a more positive light.

---

> > ### Comment · Reviewer_xjD7 · 2025-06-09
> > **Reviewer Response**
> >
> > Thank you for your responses! I have increased my score.
> >
> > One comment:
> >
> > > However, in Figure 3, we reported scores from a single run where we achieved 63.5 pass@1. This was done for consistency, as we used a single-run evaluation for checkpoints trained with 25k, 50k, 100k, 200k, and 400k samples.
> >
> > This justification makes sense to me, but I think it would improve the clarity of this paper to state this choice and note differences explicitly e.g. by explaining this in the Figure caption and/or by adding an asterisk to the result.

---

> > > ### Author Response · Authors · 2025-06-09
> > >
> > > Thank you for a positive consideration of our work. We will definitely make it explicit in the figure that the numbers are from one evaluation run.

---

> ### Author Response · Authors · 2025-06-06
> **Seeking feedback**
>
> Dear Reviewer,
>
> We addressed the concerns you raised during the rebuttal phase. If anything remains unclear, please let us know—we would be glad to provide additional information or run quick experiments before the discussion period ends. If our clarifications resolve your concerns, we would greatly appreciate your reconsideration of the current score.
>
> Thank you again for your thoughtful feedback.

---

### Official Review · Reviewer_dr9w · 2025-05-12

**Rating:** 7
**Confidence:** 4
**Ethics Flag:** 1

**Summary:**

This paper presents OpenCodeReasoning -- a synthetic coding dataset focused on reasoning. OpenCodeReasoning is created by sampling the DeepSeek-R1 model on competitive programming problems to generate solutions as well as reasoning traces. The authors conducted extensive evaluation and demonstrated that the distilled model can achieve better results on similar sized SFT-only models on coding benchmarks.

**Questions To Authors:**

Please address the concerns I had in the reasons to reject section

**Reasons To Accept:**

- This work tackles an important problem of improving code generation capabilities of models
- The introduction of a new SFT training dataset can be useful to further improve/evaluate future work
- The evaluation performed in this work is extensive and showcases the performance
- The authors provided very clear instructions and outlines on how the dataset is constructed

**Reasons To Reject:**

Fairness of the baseline comparison:
- In Section 3.1 the authors compared the distilled models fromusing OpenCodeReasoning against other distilled models using different datasets
- I am not sure how reasonable this is, since the authors only evaluate on the two benchmarks: LiveCodeBench and CodeContest both of which are very similar (i.e., competitive coding questions) to the OCR dataset.
- Other distilled models might not be distilled only for the specific task of competitive coding

The quality of the distilled dataset:
- From reading section 4.1 about the ablation of the correctness of code execution, it seems that incorrect solutions are actually better for training according to table 3?
- This seems very counter intutive to what I would expect, is this due to some distribtuion in the reasoning traces?
- I think this is a very critcial point to evaluate more on if the authors were to publish this as a training dataset

---

> ### Author Response · Authors · 2025-05-30
> **[Author response] addressing questions and concerns**
>
> We appreciate the reviewer's feedback on our work and are happy to address the questions and concerns raised.
>
> **Dataset objective and baselines**
>
> As our paper's title suggests, this work focuses on data distillation for **competitive coding**. While most baselines aren't exclusively code-only models, OlympicCoder-7B and OlympicCoder-32B are indeed code-specific. Our primary goal is to contribute a competitive coding dataset that, when combined with other datasets, can facilitate future research in developing reasoning-enabled LLMs.
>
> **Are incorrect solutions better for training?**
>
> Regarding the observation that incorrect solutions sometimes lead to better performance than correct ones, and that combining both yields the best results: Figure 4 in our paper explains why incorrect solutions can improve LLM performance on target benchmarks. For many difficult programming questions, DeepSeek-R1 couldn't generate correct solutions. Nevertheless, data distillation using these incorrect solutions still improved performance on medium and hard questions within the LiveCodeBench benchmark. Without distilling these incorrect solutions, LLMs wouldn't get the chance to learn from tougher questions.
>
> ---
>
> We'd be happy to address any other questions or concerns you have that could improve your perspective on our work.

---

> > ### Comment · Reviewer_dr9w · 2025-05-31
> >
> > Thanks for the response to my initial questions and concerns, I have increased my score

---

> > > ### Author Response · Authors · 2025-06-01
> > >
> > > Thank you for the considerations.

---

### Decision · Program_Chairs · 2025-07-08

**Decision:**

Accept

**Comment:**

This paper presents OpenCodeReasoning -- a synthetic coding dataset focused on reasoning. OpenCodeReasoning is created by sampling the DeepSeek-R1 model on competitive programming problems to generate solutions as well as reasoning traces. The authors conducted extensive evaluation and demonstrated that the distilled model can achieve better results on similar sized SFT-only models on coding benchmarks.

Reasons To Accept:
- This paper can be a valuable community resource. As far as I know, it is the largest collection of reasoning traces and solutions to programming problems to date.
- The dataset can be used to train interesting future models. It also facilitates future analysis.
- The authors provided very clear instructions and outlines on how the dataset is constructed
- Training SOTA open-weight model using this data, demonstrating its effectiveness.

Reasons To Reject:
Reviewers raised a few issues that have mostly been addressed during the discussion period, including thoughts about fair baselines, quality of the dataset, and error bars.
- One reviewer raised the issue of limited scope (focusing only on competitive programming), but this is not enough reason for rejection because significant literature focuses solely on competitive programming problems.